# GD-COMET: A Geo-Diverse Commonsense Inference Model

**Mehar Bhatia** and **Vered Shwartz**
University of British Columbia
Vector Institute for AI
{meharb23, vshwartz}@cs.ubc.ca

## Abstract

With the increasing integration of AI into everyday life, it's becoming crucial to design AI systems that serve users from diverse backgrounds by making them culturally aware. In this paper, we present GD-COMET, a geo-diverse version of the COMET commonsense inference model. GD-COMET goes beyond Western commonsense knowledge and is capable of generating inferences pertaining to a broad range of cultures. We demonstrate the effectiveness of GD-COMET through a comprehensive human evaluation across 5 diverse cultures, as well as extrinsic evaluation on a geo-diverse task. The evaluation shows that GD-COMET captures and generates culturally nuanced commonsense knowledge, demonstrating its potential to benefit NLP applications across the board and contribute to making NLP more inclusive.

## 1 Introduction

Culture plays a significant role in shaping an individual's worldviews, beliefs, behaviours, and communication styles (Spradley, 1987). A considerable portion of what is commonly referred to as commonsense knowledge is not universal but rather culture-specific, including social norms, values, traditions, and more. An example of cultural differences is greetings, which may involve a handshake in Western cultures, bowing in some Asian cultures, a 'namaste' gesture in India, or 'wai' in Thailand.

With AI systems becoming increasingly ubiquitous in society, it is imperative to go beyond the Western cultural perspective (Hershcovich et al., 2022). Lack of cultural awareness may lead to models perpetuating stereotypes and reinforcing societal inequalities (Hutchinson et al., 2020; Ross et al., 2021; Søgaard, 2022), impeding their effectiveness for users from non-Western countries.

In this paper, we focus on a popular model for commonsense reasoning, COMET (Bosselut et al., 2019), which is based on an English language

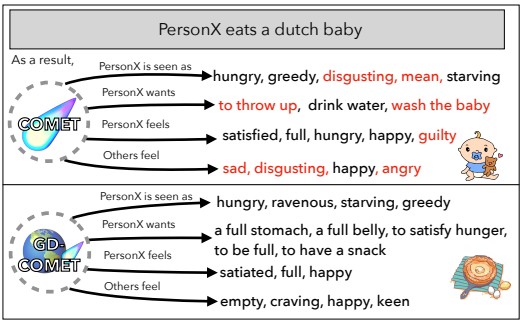

Figure 1: Inferences from COMET and GD-COMET for the sentence "PersonX eats a dutch baby", demonstrating lack of culture awareness in COMET.

model (LM) and further trained on commonsense inferences collected from North American crowd-source workers (Sap et al., 2019). Consequently, the model exhibits a certain bias towards the North American cultural perspective. As evidenced by Fig. 1, COMET displays limited familiarity with the concept of a German pancake, erroneously interpreting the term "dutch baby" in a literal sense.

We identify a need for more inclusive commonsense reasoning models and propose GD-COMET: **G**eo-**D**iverse COMET. As demonstrated in Fig 1, GD-COMET gained the culturally relevant knowledge to interpret "dutch baby" as a legitimate dish.

GD-COMET is similarly based on an English LM but is trained on a knowledge base of cultural knowledge (Nguyen et al., 2023) prior to training on COMET's original training data. This simple approach is effective, as judged by both human evaluations as well as extrinsic evaluation on a geo-diverse task (Yin et al., 2021). GD-COMET can potentially benefit many downstream NLP applications where the user population is diverse.[1]

## 2 Background

### 2.1 Commonsense Inference Models

Many NLP tasks require reasoning beyond what is explicitly stated in the text. People fill in those gaps

---

[1] Code available at github.com/meharbhatia/GD-COMET

with their commonsense knowledge. NLP models attempt to do the same by leveraging commonsense knowledge bases (KBs) such as ConceptNet (Speer et al., 2017) and ATOMIC (Sap et al., 2019). To achieve better coverage, knowledge models such as COMET (Bosselut et al., 2019) are based on pre-trained LMs and further fine-tuned on KBs, enabling contextually-relevant inferences along the KB's dimensions for new contexts.

COMET's hybrid approach proved useful for various tasks (e.g., Chakrabarty et al., 2020; Ammanabrolu et al., 2021; Ravi et al., 2023a). Subsequent versions of COMET have been developed to draw inferences from paragraphs (Gabriel et al., 2021), images (Park et al., 2020), and complex sentences (Ravi et al., 2023b). Further improvements include obtaining additional training data through crowdsourcing (Hwang et al., 2021) or generating synthetic data from LMs (West et al., 2022).

COMET and its successors assume the universality of commonsense knowledge, yet much of this knowledge may differ among cultures, in traditions (e.g., duration of a wedding ceremony; Acharya et al., 2021), foods (e.g., what counts as breakfast food; Speer et al., 2017), social norms, and more.

## 2.2 Culture-Aware NLP

While multilingual NLP is a popular topic, culture-aware NLP is under-explored. It is crucial for language technologies to not only serve speakers of a wide variety of languages but also acknowledge that users come from diverse cultures (Hershcovich et al., 2022). Cultural norms and pragmatic aspects differ across speakers from different cultures (Zhou et al., 2023). Nevertheless, English LMs primarily reflect a North-American lens due to training on web data with a US user bias (Cao et al., 2023).

Current work in culture-aware NLP addresses various aspects. One line of work focuses on cultural stereotypes and biases, and ways to measure and mitigate them (e.g., Hutchinson et al., 2020; Ross et al., 2021; Søgaard, 2022). Another line of work analyzes the differences in culture-specific commonsense knowledge, including relational knowledge (Yin et al., 2022), grounding of time expressions (Shwartz, 2022), food-related customs (Palta and Rudinger, 2023) and social values (Lin et al., 2021; Arora et al., 2023). At the same time, there have been efforts to develop benchmarks (Yin et al., 2021; Liu et al., 2021), and adapt models to new cultures (Zhou et al., 2023; Yin et al.,

2023). Finally, there are several recent cultural KBs such as StereoKG (Deshpande et al., 2022), Quasimodo (Romero et al., 2019), and CANDLE (Nguyen et al., 2023). CANDLE, which we use in this work, is the most comprehensive among them, containing 1.1M assertions in English about 386 cultures (e.g. "A Dutch baby is a German pancake that is baked instead of cooked on the stove top"). CANDLE assertions were extracted from a large web corpus and clustered into *5 facets of culture*: food, drinks, clothing, rituals, and traditions.

## 3 GD-COMET

We present GD-COMET, a geo-diverse version of COMET. The goal of GD-COMET is to generate high-quality commonsense inferences for concepts and events pertaining to both Western and non-Western cultures. Rather than collecting a large-scale geo-diverse dataset in the style of ATOMIC, we split the training into two phases: (1) training the underlying LM on geo-diverse data; (2) continue training on the large-scale original COMET training data. This is motivated by Bosselut et al. (2019) that showed that implicit commonsense knowledge from underlying LM's pre-training transfers to COMET. We similarly hypothesize that encoding geo-diverse data into the underlying LM prior to training on COMET data will transfer this knowledge to GD-COMET.

**Geo-Diverse Training (GD-BART).** We pick 770,000 assertions from CANDLE with a combined score greater than 0.5. This threshold selects highly distinctive assertions specific and relevant to the specific region. We fine-tune BART-Large, the underlying LM of the latest COMET model (Hwang et al., 2021), on this data, using BART's original pre-training objectives (token masking, token deletion, text infilling and sentence permutation). We save the model checkpoint with the lowest validation loss after training for 50 epochs on two NVIDIA A40 GPUs.

**COMET Training.** We proceed to fine-tuning GD-BART on the large-scale ATOMIC-2020 dataset, using the same training method and hyperparameters as Hwang et al. (2021). Appendix A lists the 34 COMET relations used in this paper.

## 4 Intrinsic Evaluation

To evaluate the quality of GD-COMET, we construct a set of input sentences pertaining to 5 diverse cul-

| | | ① | ② | ③ | Average $\kappa$ |
|---|---|---|---|---|---|
| COMET | India | 2.32 | 2.16 | 2.65 | 0.71 |
| | S Korea | 1.93 | 1.86 | 2.32 | 0.67 |
| | Nigeria | 1.97 | 1.98 | 2.27 | 0.61 |
| | Iran | 2.09 | 2.31 | 2.42 | 0.63 |
| | Indonesia | 2.28 | 2.36 | 2.55 | 0.66 |
| | | ① | ② | ③ | Average $\kappa$ |
| GD-COMET | India | 2.62 | 2.54 | 2.73 | 0.74 |
| | S Korea | 2.13 | 1.92 | 2.35 | 0.65 |
| | Nigeria | 2.25 | 1.92 | 2.35 | 0.59 |
| | Iran | 2.27 | 2.38 | 2.58 | 0.76 |
| | Indonesia | 2.43 | 2.46 | 2.58 | 0.77 |

Table 1: Evaluation of COMET and GD-COMET inferences, judged by annotators from the respective cultures.

tures (Table 1). We sample 5 concepts for each facet and use facet-specific templates (Appendix B) to create 20 sentences for each culture. For each of COMET and GD-COMET, we use beam search to generate 5 inferences for each of the 34 dimensions and convert them to natural language statements using relation-specific templates based on prior work (Bosselut et al., 2019). The correctness of both inferences were judged by 10 graduate students, two students from each of the respective cultures. Annotators were asked to grade inferences along the following criteria on scale of 0 (worst) to 3 (best):

① **Cultural Relevance:** The inference is factually accurate and reflects the values, customs, traditions, and societal norms associated with the given culture.

② **Stereotype Avoidance:** The inference does not perpetuate stereotypes about the culture.

③ **Linguistic Accuracy:** The inference is grammatical, and the vocabulary and idiomatic expressions are appropriate in that culture.

The annotations yielded a substantial inter-annotator agreement with $\kappa = 0.656$ for COMET and 0.702 for GD-COMET, measured with average Cohen's Kappa (Cohen, 1960) across cultures.

**Results.** Table 1 reveals that GD-COMET consistently outperforms the standard COMET model. Specifically, GD-COMET excels in generating culturally aligned inferences across chosen diverse cultures, and is more likely than COMET to avoid biased assumptions. However, there is still room for improvement for South Korea and Nigeria.

## 5 Extrinsic Evaluation

Traditional benchmarks often fall short in testing models' knowledge and comprehension of diverse cultural contexts. To show GD-COMET's utility for downstream tasks, we evaluate on a multimodal task, GD-VCR (Sec 5.1). We develop a model inspired by VLC-BERT (Ravi et al., 2023a) that generates inferences and incorporates them into a vision and language (V&L) model (Sec 5.2). We show that GD-COMET improves the performance on GD-VCR upon an array of baselines (Sec 5.3) and demonstrate the inferences contributing to the performance gains (Sec 5.4).

### 5.1 Dataset

Visual Commonsense Reasoning (VCR; Zellers et al., 2019) is a benchmark for testing V&L models' ability to understand and reason beyond a visual scene. Each example consists of an image extracted from movies or TV series and a multiple-choice question about the actions or people depicted in the image. This dataset focuses solely on Western, primarily North American movies.

Geo-Diverse Visual Commonsense Reasoning dataset (GD-VCR; Yin et al., 2021) follows the same setup of VCR but extends to diverse regions. This evaluation-only dataset includes 328 images from movies and TV series in East Asian, South Asian, African and Western countries (See Appendix C). We follow the original setup and train our model on VCR before testing on GD-VCR.

### 5.2 Model (VLC-BERT with GD-COMET)

We take inspiration from VLC-BERT (Ravi et al., 2023a), that incorporated COMET inferences into VL-BERT (Su et al., 2019). Instead, we integrate GD-COMET as a source of contextualized cultural commonsense knowledge for GD-VCR. Figure 2 illustrates the model. We describe below VLC-BERT and where our model deviates from it.

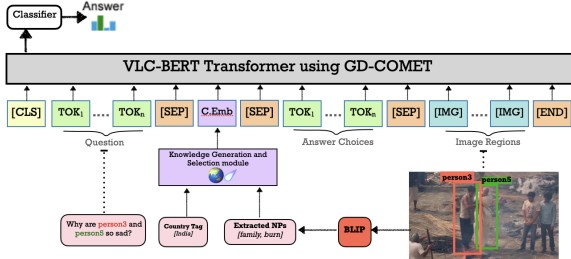

Figure 2: A model using GD-COMET for GD-VCR.

| Datasets | Human | VisualBERT* | ViLBERT* | VL-BERT | VLC-BERT w/ | | |
|---|---|---|---|---|---|---|---|
| | | | | | GD-BART | COMET | GD-COMET |
| **GD-VCR** | 88.84 | 53.27 | 58.47 | 58.63 | 52.69 | 59.59 | **63.51** |
| ○ **West** | 91.23 | 65.82 | 64.37 | 65.27 | 57.69 | 66.78 | **69.93** |
| ○ **South Asia** | 92.98 | 52.04 | 62.9 | 64.92 | 54.35 | 64.25 | **68.17** |
| ○ **Africa** | 87.93 | 51.85 | 62.04 | 58.17 | 51.87 | 57.71 | **64.81** |
| ○ **East Asia** | 83.05 | 45.39 | 46.45 | 47.88 | 41.87 | 49.64 | **53.07** |

Table 2: Accuracy (%) of the different models on the subset of each region in GD-VCR. We report the average across 3 runs (see Appendix D for the results of individual seeds). Results marked with ∗ have been reported in Yin et al. (2021).

**Knowledge Generation and Selection.** VLC-BERT uses the question and the object tags as input to COMET. Instead of object tags, we generate an image caption using BLIP (Li et al., 2023) and extract noun phrases from the caption using SpaCy (Honnibal et al., 2020). We found that the noun phrases provide a more detailed description of the depicted activities within the image (e.g. "family, burn" in Fig. 2). We additionally append a country tag to the input. During training on VCR, we use the tag "North America", the primary source of movies in the dataset. For the images in GD-VCR, we extracted country tags from Wikipedia.

We use beam search to generate five inferences for each of the 34 dimensions. To select the most relevant inferences, we convert the inferences to natural language sentences using relation-specific templates and select the inferences that are the most similar to the question using SBERT embeddings (Reimers and Gurevych, 2019).

**Overall Architecture.** The generic input to VL-BERT for VCR is <question, answer tokens, image regions>. Following Ravi et al. (2023a), we embed each inference with SBERT and summarize them into a single token with a weighted average based on learned attention scores. Finally, we feed the output of the [CLS] token into a classifier to predict a score for each answer choice. We train the model using binary cross-entropy loss for 20 epochs on 4 NVIDIA RTX6000 GPUs.

### 5.3 Results

Table 2 compares our model's performance on GD-VCR with baselines that: (i) do not make use of commonsense knowledge (VL-BERT); (ii) generate inferences using GD-BART; and (iii) use COMET (VLC-BERT w/COMET). Note that the same signals (i.e., country tag and noun phrases) were used for the GD-BART and COMET baselines. We also include prior results reported using VisualBERT and ViLBERT for completeness.

VLC-BERT w/COMET modestly improves upon VL-BERT across most regions, with an overall improvement of 1.2 points in accuracy. This suggests that COMET provides some commonsense inferences that are universal. Conversely, GD-COMET shows a substantial improvement of nearly 5 points over VL-BERT and 4 points over VLC-BERT w/COMET. This highlights the effectiveness of incorporating GD-COMET for downstream tasks that require culture-specific knowledge across diverse regions. Furthermore, GD-BART performs less effectively than other methods, underscoring the importance of training on structured knowledge to generate contextually relevant responses.

### 5.4 Qualitative Analysis

Figures 3 presents several GD-VCR instances along with the models' predictions, and the inferences generated by COMET and GD-COMET for them. In Figure 3a, GD-COMET accurately associates a girl wearing henna in Somalia with marriage. In Figure 3b, it understands that folding palms during an Indian festival signifies a greeting or welcome. Finally, in Figure 3c, it recognizes that bowing in South Korea is a gesture of apology, leading to VLC-BERT w/ GD-COMET to be the only model that provides a correct answer. In contrast, COMET's inferences for this example are generic and irrelevant. These examples highlight GD-COMET's effectiveness in identifying the cultural context and dynamically generating culturally-relevant commonsense inferences across ATOMIC's relations.

## 6 Conclusion

This work challenges the current notion of universally applicable commonsense knowledge by introducing GD-COMET, a geo-diverse variant of COMET. GD-COMET can generate culturally-nuanced commonsense inferences for a broad range of cultures. Our comprehensive evaluation con-

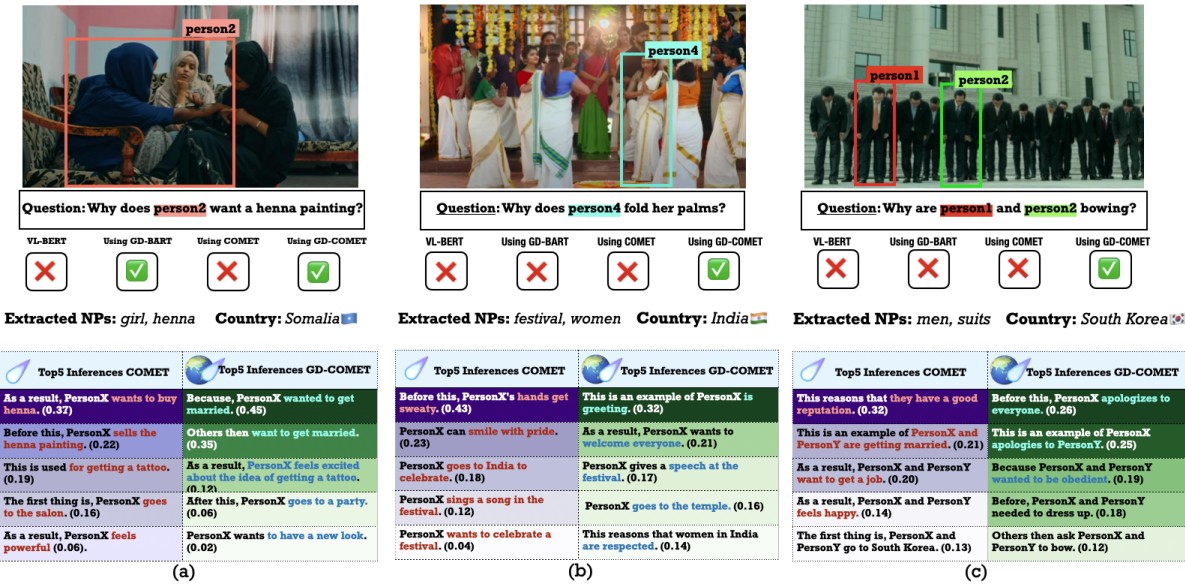

Figure 3: Attention analysis of commonsense inferences generated by COMET and GD-COMET for testing samples in GD-VCR.

firms the effectiveness of GD-COMET in incorporating and leveraging cultural cues. We view our work as a step towards developing more inclusive and culturally-aware AI systems.

## Limitations

While GD-COMET represents a significant advancement in incorporating cultural commonsense knowledge into AI models, a few limitations need to be acknowledged.

First, the availability of comprehensive, high-quality data remains a challenge in training culturally-aware models. While resources like CANDLE provide a step forward in curating diverse cultural knowledge, it is essential to note that merely capturing the existence of concepts within a culture is insufficient. Future efforts should aim to collect data that reflects the presence of certain concepts and encompasses how people perceive and interpret those concepts within their specific cultural contexts. This would require extensive data collection efforts that go beyond surface-level understanding, and delve into the nuances of cultural perspectives.

A second limitation is the availability of suitable benchmarks for testing models' knowledge and understanding of cultural variations. In particular, two such tasks, GD-VCR and MarVL (Liu et al., 2021), focus on vision and language, while Nguyen et al. (2023) proposes a cultural knowledge quiz. We hope to see more language-only datasets developed to go beyond testing models on knowledge about concepts from diverse cultures to understanding cultural nuances.

## Ethics Statement

Despite being designed to be more culturally inclusive, GD-COMET runs the risk of unintentionally perpetuating biases present in CANDLE data. In particular, CANDLE might misrepresent cultures with stereotypes or underrepresent cultures. Addressing these concerns requires proactive measures such as identifying biases using methods such as Mehrabi et al. (2021) and mitigating them through filtering and additional data collection.

Additionally, the size of evaluation benchmarks means they don't always account for cultural variations within the same region. For example, GD-VCR images in the African region are concentrated in East Africa. Similarly, addressing this issue would require additional annotation efforts.

## Acknowledgement

This work was funded, in part, by the Vector Institute for AI, Canada CIFAR AI Chairs program, an NSERC discovery grant, and a research gift from AI2. Finally, we sincerely thank Sahithya Ravi, Aditya Chinchure, Ward Pennink and Jan Zimny for valuable feedback and discussions.

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

## A   COMET Relations

Table 3 lists COMET relations used in this work.

| | | |
|---|---|---|
| AtLocation | CapableOf | isBefore |
| Causes | CausesDesire | isFilledBy |
| CreatedBy | Desires | oEffect |
| HasPrerequisite | HasFirstSubevent | oReact |
| HasA | HasProperty | oWant |
| InstanceOf | IsA | xAttr |
| LocatedNear | MadeOf | xEffect |
| MadeUpOf | MotivatedByGoal | xIntent |
| ObjectUse | PartOf | xNeed |
| ReceivesAction | SymbolOf | xReact |
| UsedFor | isAfter | xReason |
| xWant | | |

Table 3: COMET relations used in this work.

## B   Facet Templates

Table 4 presents the templates used for creating input sentences to GD-COMET for each concept associated with a cultural facet.

| | |
|---|---|
| **clothing** | "PersonX wears [concept] in [country]" |
| **food** | "PersonX eats [concept] in [country]" |
| **drink** | "PersonX drinks [concept] in [country]" |
| **festival** | "PersonX celebrates [concept] in [country]" |

Table 4: Templates used to create input sentences for GD-COMET for each CANDLE facet.

## C   VCR and GD-VCR Statistics

Table 5 displays the statistics of the VCR and GD-VCR datasets. The bottom half shows the number of images for each region in GD-VCR.

| | # Images | # QA Pairs | avg Q length | avg A length |
|---|---|---|---|---|
| **VCR (dev)** | 9929 | 26534 | 6.77 | 7.67 |
| **GD-VCR** | 328 | 886 | 7.38 | 7.68 |
| **West** | 100 | 275 | 7.36 | 7.19 |
| **East Asia** | 101 | 282 | 7.59 | 7.59 |
| **South Asia** | 87 | 221 | 6.85 | 8.00 |
| **Africa** | 40 | 108 | 7.98 | 8.54 |

Table 5: Statistics of the VCR and GD-VCR benchmarks.

## D   Full Performance

Extending upon Table 2, we provide a complete summary of the results of individual seeds on GD-VCR in Table 6.

| Models | Overall | West | South Asia | East Asia | Africa |
|---|---|---|---|---|---|
| *Human Performance* | *88.84* | *91.23* | *92.98* | *83.05* | *87.93* |
| VisualBERT* | 53.27 | 62.91 | 52.04 | 45.39 | 51.85 |
| ViLBERT* | 58.47 | 65.82 | 62.9 | 46.45 | 62.04 |
| VL-BERT *(seed 1)* | 58.8 | 64.73 | 67.42 | 48.58 | 54.78 |
| VL-BERT *(seed 2)* | 58.92 | 65.82 | 63.8 | 47.16 | 62.04 |
| VL-BERT *(seed 3)* | 58.19 | 65.27 | 63.54 | 47.91 | 59.7 |
| VL-BERT *(average)* | 58.63 | 65.27 | 64.92 | 47.88 | 58.17 |
| VLC-BERT with GD-BART *(seed 1)* | 53.63 | 58.54 | 54.08 | 41.42 | 52.78 |
| VLC-BERT with GD-BART *(seed 2)* | 52.92 | 57.27 | 54.08 | 42.09 | 52.13 |
| VLC-BERT with GD-BART *(seed 3)* | 51.55 | 57.27 | 54.89 | 42.09 | 50.70 |
| VLC-BERT with GD-BART *(average)* | 52.69 | 57.69 | 54.35 | 41.87 | 51.87 |
| VLC-BERT with COMET *(seed 1)* | 59.71 | 67.27 | 64.71 | 50.00 | 55.56 |
| VLC-BERT with COMET *(seed 2)* | 59.82 | 67.27 | 64.71 | 50.36 | 55.56 |
| VLC-BERT with COMET *(seed 3)* | 59.25 | 65.82 | 63.35 | 48.58 | 62.04 |
| VLC-BERT with COMET *(average)* | 59.59 | 66.78 | 64.25 | 49.64 | 57.71 |
| VLC-BERT with GD-COMET *(seed1)* | 62.87 | 67.64 | 70.14 | 51.77 | 64.81 |
| VLC-BERT with GD-COMET *(seed2)* | 65.01 | 72.36 | 67.42 | 54.97 | 67.59 |
| VLC-BERT with GD-COMET *(seed3)* | 62.64 | 69.82 | 66.97 | 52.48 | 62.03 |
| **VLC-BERT with GD-COMET *(average)*** | **63.51** | **69.93** | **68.17** | **53.07** | **64.81** |

Table 6: Accuracy (%) of the various models on the subset of each region in GD-VCR. Results marked with ∗ have been reported in Yin et al. (2021).