# OpenReview forum: "GD-COMET: A Geo-Diverse Commonsense Inference Model"
_EMNLP/2023/Conference — EMNLP 2023 Main_

### Official Review · Reviewer_3GPm · 2023-07-28

**Soundness:** 3

**Excitement:**

3: Ambivalent: It has merits (e.g., it reports state-of-the-art results, the idea is nice), but there are key weaknesses (e.g., it describes incremental work), and it can significantly benefit from another round of revision. However, I won't object to accepting it if my co-reviewers champion it.

**Paper Topic And Main Contributions:**

This paper addresses the issue that the commonsense inference model COMET was trained on English data based on Western culture, which limits its capability to perform inference accurately for other cultures. To overcome this, the authors propose to finetune the underlying model of COMET (BART large) on the CANDLE dataset before finetuning it on the ATOMIC dataset, resulting in their GD-COMET model that can leverage geo-diverse knowledge. They performed a human evaluation on GD-COMET-generated inference statements and also conducted experiments to show that GD-COMET could be helpful on a GD visual commonsense reasoning task, by incorporating GD-COMET-generated inference into the input of a VLC-BERT model.

**Questions For The Authors:**

1.  In Section 4, could you provide some examples of the inference statements used for annotation and explain in more detail how the "culture relevance" is scored? It is not clear in the paper whether the relevance score indicates correctness or just topical relevance.

2. What is your intuition regarding the difference in scores among the selected cultures?


**Reasons To Accept:**

1. The paper focus on an interesting problem, that commonsense knowledge should be cultural-specific.They proposed a simple solution to mitigate the problem, which is to pre-train the common sense inference model on geo-diverse statements.
2. The problem statement and description of their method are clear.


**Reasons To Reject:**

1. I think the proposed mitigating method (pretraining on CANDLE data) is simple, which is ok, but there is a lack of analysis regarding how and why pretraining on CANDLE contributed to the geo-diverse commonsense inference.

2. The evaluation conducted in the paper is not entirely convincing:
(1) The intrinsic evaluation section lacks a comparison of how the baseline models performed, making it difficult to determine whether GD-COMET improved cultural-specific inference.
(2)  The extrinsic evaluation looks more sound, but it is unclear whether the same signals were used for the VLC-BERT w GD-BART or COMET baselines. For example, the authors introduced additional signals such as country tag and extracted NPs when evaluating with GD-COMET, and it was not clear if the same signals have been used for the other two baselines.
(3) I think in general the paper could benefit from evaluating on more datasets if possible, as the dataset used in the extrinsic evaluation is small.

3. The paper does not provide an explanation for the difference in model performance across the selected cultures in both human and downstream task evaluations.


**Reproducibility:**

4: Could mostly reproduce the results, but there may be some variation because of sample variance or minor variations in their interpretation of the protocol or method.

**Reviewer Confidence:**

4: Quite sure. I tried to check the important points carefully. It's unlikely, though conceivable, that I missed something that should affect my ratings.

---

> ### Author Rebuttal · Authors · 2023-08-28
>
> We are grateful for the detailed and thorough review of our paper and will use the constructive feedback to enhance the quality of our work. Below, we provide our responses to your comments:
>
> **In Section 4, could you provide some examples of the inference statements used for annotation and explain in more detail how the "culture relevance" is scored? It is not clear in the paper whether the relevance score indicates correctness or just topical relevance.**
>
> The "culture relevance" score aims to assess the extent to which the inference generated by our model aligns with the values, customs, traditions, and societal norms associated with a specific culture. In essence, it is correct to consider both correctness (i.e., being factually accurate) and topical relevance (i.e., pertinent to the culture's themes, practices, or historical background). We will add the specific instructions and examples that were provided to the annotators to the appendix.
>
> **I think the proposed mitigating method (pretraining on CANDLE data) is simple, which is ok, but there is a lack of analysis regarding how and why pretraining on CANDLE contributed to the geo-diverse commonsense inference.**
> **(1) The intrinsic evaluation section lacks a comparison of how the baseline models performed, making it difficult to determine whether GD-COMET improved cultural-specific inference.**
>
> As also requested by Reviewer 1, we repeated the intrinsic evaluation using the original COMET model. Across all five cultures, we see average scores of 2.118, 2.134, and 2.442  for our three metrics (compared to average scores of 2.34, 2.244, and 2.518 of GD-COMET assertions), supporting our claim that GD-COMET demonstrates improved cultural-specific assertions compared to vanilla COMET. We will update the paper with the results. We also believe that this evaluation provides additional empirical evidence on how CANDLE contributes to generating culturally-aligned inferences.
>
> **(2) The extrinsic evaluation looks more sound, but it is unclear whether the same signals were used for the VLC-BERT w GD-BART or COMET baselines.**
>
> Yes, We used the same inputs, i.e. country tag and noun phrases, for training both other baselines.  We will clarify this in the paper.
>
> **(3) I think, in general, the paper could benefit from evaluating more datasets if possible, as the dataset used in the extrinsic evaluation is small.**
>
> As we mention in the limitation section, there are very few appropriate benchmarks for assessing models' cultural knowledge and comprehension.  However,  given that this topic is now gaining popularity and new datasets are being released, we plan to test GD-COMET on new datasets such as Fork (Palta and Rudinger, 2023) in the future.
>
> **What is your intuition regarding the difference in scores among the selected cultures?**
>
> We appreciate the insightful question about the observed score variations among the selected cultures in our evaluation. As can be seen by comparing the results of VL-BERT in Table 2, the improvement from GD-COMET is relatively uniform across cultures, suggesting that the gap originates in the underlying LM (i.e. BERT) itself. Understanding why models represent some cultures better than others is an important question that we plan to address in future work. We envisage that there are several factors, such as the availability of cultural data, cultural complexity and diversity.

---

### Official Review · Reviewer_J9XN · 2023-07-31

**Soundness:** 4

**Excitement:**

4: Strong: This paper deepens the understanding of some phenomenon or lowers the barriers to an existing research direction.

**Missing References:**

FORK: A Bite-Sized Test Set for Probing Culinary Cultural Biases in Commonsense Reasoning Models
Shramay Palta, and Rachel Rudinger
Findings of the Association for Computational Linguistics: ACL 2023

(might be concurrent work, but in any case would be good to cite it for the camera-ready)

**Paper Topic And Main Contributions:**

This paper trains a culturally-aware commonsense inference model. It does so by first training a LM on geo-diverse data, which it obtained from CANDLE (a cultural KB). The model then gets further trained on COMET training data. The paper evaluates their model intrinsically using human evaluation and extrinsically on a visual commonsense reasoning. In both cases they show that their model produces more culturally aligned outputs compared to other approaches.

**Questions For The Authors:**

Lines 134-137: Do you think it makes a difference whether you first train on 1 vs. 2?
Lines 145-156: Do you specify the region when fine-tuning?
Lines 165-168: will you publish the evaluation set?


**Reasons To Accept:**

This paper discusses an important, and often neglected aspect of NLP: how to to train culturally pluralistic models. It offers an interesting approach towards solving this issue and convincingly evaluates the model on intrinsic and extrinsic tasks. The evaluation seems to be well thought out and they also make sure to check whether the inference does perpetuate stereotypes about a given culture.

**Reasons To Reject:**

Besides the clarification questions asked below I don’t see any obvious risks with having this paper accepted. For future work it would be interesting to see an analysis on why the model does better on certain cultures than others.

**Reproducibility:**

5: Could easily reproduce the results.

**Reviewer Confidence:**

4: Quite sure. I tried to check the important points carefully. It's unlikely, though conceivable, that I missed something that should affect my ratings.

---

> ### Author Rebuttal · Authors · 2023-08-28
>
> We sincerely appreciate the thorough review and valuable feedback, and we are grateful for your recognition of the importance of training culturally aware NLP models. We have carefully considered each of the points raised and have made corresponding adjustments to enhance the quality and clarity of our paper. Below, we address the reviewer's comments:
>
> **Lines 134-137: Do you think it makes a difference whether you first train on 1 vs. 2?**
>
> This is an interesting question.  We did not study the effect of reversing the order of training phases since we hypothesized that if we train on ATOMIC-style data first as the first step and later on CANDLE’s unstructured data, we might lose the ability to generate commonsense inferences along ATOMIC’s relations. In the future, we will explore more advanced scheduling and multi-task training techniques.
>
> **Lines 145-156: Do you specify the region when fine-tuning?**
>
> We do not explicitly add a region tag again when fine-tuning BART with CANDLE data. The reason is that most sentences in CANDLE data already contain the region name (e.g. “Paneer is a type of cheese that is popular in Indian cuisine”). We will clarify this and include some examples from CANDLE in the camera-copy version.
>
> **Lines 165-168: Will you publish the evaluation set?**
>
> Yes!
>
> **Missing References**
>
> Thank you for the missing reference! We became aware of FORK at ACL’23 and will make sure to cite it in the final version.

---

### Official Review · Reviewer_La5x · 2023-08-08

**Typos Grammar Style And Presentation Improvements:** 1. Figures are blurry. Visual effects…
**Soundness:** 4

**Excitement:**

4: Strong: This paper deepens the understanding of some phenomenon or lowers the barriers to an existing research direction.

**Paper Topic And Main Contributions:**

This paper presents a way to improve commonsense knowledge models, such as COMET, into a Geo-diverse aware model.

The author pretrains the backbone LM on culture commonsense assertions and then trains on ATOMIC-2020 as the original COMET did. The culture commonsense originates from CANDLE, which is an existing dataset collected through web mining and consists of assertions explicitly with country names.

As for the evaluation, the author did a human evaluation to test the model's ability to generate cultural-relevant commonsense assertions. The author also tests their model with a multi-modal downstream task, which is Geo-diverse visual commonsense reasoning, using their model as the knowledge generation component.

Main Contribution:
1. The work provides a cultural-aware knowledge model that can be easily used.
2. This paper validates the intuition that pretraining LM on cultural commonsense assertions can improve the model's cultural-awareness, which is evaluated both on the assertion generation itself and a multi-modal downstream task.


**Questions For The Authors:**

A. How does the original COMET perform when adding the country information as in Section 4?  Is there a human evaluation done on that as well to show the GD-COMET generates better assertions than the baseline?

B. In section 5.4, what I've seen in CANDLE is "In Korea, the bow is the standard greeting." and nothing about apology. Is the qualitative example just the effect of the randomness of the training starting point?



**Reasons To Accept:**

1. The cultural-awareness aspect of LM is an issue that requires more collected effort of the community.  This work provide a validated useful tool for followed-up researches (We've seen how COMET is widely-used for commonsense related tasks).

2. The evaluation is comprehensive. An human evaluation is carried out, instead those meaningless automatic metrics. The model is also tested on a multimodal downstream task as the knowledge generation module, where there is a decent improvement of vanilla COMET.

3. The content is well-presented with a few illustrations that makes the paper easy to read.

**Reasons To Reject:**

1. For the intrinsic evaluation, the baseline of the original COMET is missing, which is very crucial. Otherwise, we don't have a quantitative proof (on the intrinsic side) that the model generates better cultural-aware commonsense knowledge than the original model.
2. There is a lack of analysis or intuition presented on the effect of pretraining for different COMET inputs (especially containing cultural-specific concepts or not).  To be more specific, CANDLE is mainly about noun concepts around clothing, food & drink, and some traditions. But for the GD-VCR downstream task, it is not very likely that the extracted noun phrases hit into the concepts in CANDLE, and there may need some extra explanations.


**Reproducibility:**

4: Could mostly reproduce the results, but there may be some variation because of sample variance or minor variations in their interpretation of the protocol or method.

**Reviewer Confidence:**

5: Positive that my evaluation is correct. I read the paper very carefully and I am very familiar with related work.

---

> ### Author Rebuttal · Authors · 2023-08-28
>
> We sincerely appreciate the reviewer’s time and effort in providing valuable feedback on our paper. We have carefully considered each of the comments and suggestions and have made appropriate revisions to address the concerns raised. Below, we outline our response:
>
> **Figures are blurry. Visual effects like the pencil brush and shadow make the figures a little bit more messy.**
>
> We enhanced the resolution of the figures and updated them in the paper.
>
> **At the beginning of section 5, add a short explanation of why a multi-modal setting is used as the downstream task. (Maybe move a few sentences from limitation to the main paper.)**
>
> We added a brief explanation at the beginning of Section 5 to elucidate the rationale for employing a multi-modal setting in the downstream task.
>
> **Question A. How does the original COMET perform when adding the country information as in Section 4? Is there a human evaluation done on that as well to show the GD-COMET generates better assertions than the baseline?**
>
> Thanks for pointing out the missing human evaluation of COMET. We repeated the same intrinsic evaluation for the outputs of the original COMET model. Across all five cultures, we see average scores of 2.118, 2.134, and 2.442 for our three metrics (compared to average scores of 2.34, 2.244, and 2.518 of GD-COMET assertions). This human evaluation reassesses that GD-COMET generates improved cultural-specific assertions compared to vanilla COMET. The results will be added in Section 4 in the camera-ready version.
>
> **Question B. In section 5.4, what I've seen in CANDLE is "In Korea, the bow is the standard greeting." and nothing about apology. Is the qualitative example just the effect of the randomness of the training starting point?**
>
> **But for the GD-VCR downstream task, it is not very likely that the extracted noun phrases hit into the concepts in CANDLE, and there may need some extra explanations.**
>
> We appreciate the attention to detail! We would like to clarify that while the specific example mentioned might not be explicitly present in the CANDLE dataset, it is not solely attributed to the randomness of the training starting point. Instead, it underscores one of the strengths of the GD-COMET model. Similar to COMET, GD-COMET can dynamically generate novel and relevant (and culturally aligned, unlike COMET) commonsense inferences across ATOMIC's relations. We will clarify this in the paper that, in essence, our approach encourages the model to grasp a foundational cultural understanding, allowing it to make inferences that might not be explicitly present in CANDLE.

---

### Meta-Review · Area_Chair_7yTG · 2023-09-11

**Recommendation:** 5

**Metareview:**

The authors provide a well-written paper that addresses an important issue in NLP, the cultural-awareness of commonsense knowledge models. The proposed GD-COMET, a culturally-aware generative COMET model, appears a valuable contribution to the field, and the (for a short paper) comprehensive evaluation of its benefits is a strength of the paper. There are a few areas where the paper should be further improved, such as providing a more detailed analysis of the effect of pretraining on CANDLE, and comparing the baseline models' performance in the intrinsic evaluation. Additionally, the paper could benefit from evaluating on more datasets if possible. The authors have provided detailed and helpful responses to the reviewers' comments which would allow them to address most concerns raised for the final version. Overall, this paper appears a valuable contribution to the discourse in the field.

---

### Decision · Program_Chairs · 2023-10-07

**Decision:**

Accept-Main

**Comment:**

The authors provide a well-written paper that addresses an important issue in NLP, the cultural-awareness of commonsense knowledge models. The proposed GD-COMET, a culturally-aware generative COMET model, appears a valuable contribution to the field, and the (for a short paper) comprehensive evaluation of its benefits is a strength of the paper. There are a few areas where the paper should be further improved, such as providing a more detailed analysis of the effect of pretraining on CANDLE, and comparing the baseline models' performance in the intrinsic evaluation. Additionally, the paper could benefit from evaluating on more datasets if possible. The authors have provided detailed and helpful responses to the reviewers' comments which would allow them to address most concerns raised for the final version. Overall, this paper appears a valuable contribution to the discourse in the field.